# *Morinda citrifolia* Essential Oil: A Plant Resistance Biostimulant and a Sustainable Alternative for Controlling Phytopathogens and Insect Pests

**DOI:** 10.3390/biology13070479

**Published:** 2024-06-27

**Authors:** Bruna Leticia Dias, Renato Almeida Sarmento, Madelaine Venzon, Luis Oswaldo Viteri Jumbo, Lucas Samuel Soares dos Santos, Wellington de Souza Moura, Dalmarcia de Souza Carlos Mourão, Paulo Ricardo de Sena Fernandes, Taila Renata Neitzke, João Victor de Almeida Oliveira, Tiago Dias, Mateus Sunti Dalcin, Eugênio E. Oliveira, Gil Rodrigues dos Santos

**Affiliations:** 1Programa de Pós-Graduação em Biodiversidade e Biotecnologia- Rede Bionorte, Universidade Federal do Tocantins (UFT), Gurupi 77402-970, TO, Brazil; rsarmento@uft.edu.br (R.A.S.); bussund@gmail.com (W.d.S.M.); 2Departamento de Fitopatologia, Universidade Federal do Tocantins (UFT), Gurupi 77402-970, TO, Brazil; dalmarciaadm@uft.edu.br (D.d.S.C.M.); pauloricardosena@mail.uft.edu.br (P.R.d.S.F.); tailaneitzke@gmail.com (T.R.N.); oliveira.victor@mail.uft.edu.br (J.V.d.A.O.); 3Programa de Pós-Graduação em Produção Vegetal, Universidade Federal do Tocantins, Gurupi 77402-970, TO, Brazil; mateussuntidalcin@hotmail.com; 4Agriculture and Livestock Research Enterprise of Minas Gerais (EPAMIG), Viçosa 36571-000, MG, Brazil; madelainevenzon@gmail.com; 5Programa de Pós-Graduação Ciências Florestais e Ambientais, Universidade Federal do Tocantins (UFT), Gurupi 77402-970, TO, Brazil; luis.viteri@mail.uft.edu.br; 6Programa de Pós-Graduação em Biotecnologia, Universidade Federal do Tocantins (UFT), Gurupi 77402-970, TO, Brazil; 7Departamento de Química, Universidade Federal do Tocantins, Curso de Química Ambiental, Câmpus de Gurupi, P.O. Box 66, Gurupi 77410-530, TO, Brazil; lsantos@mail.uft.edu.br; 8Departamento de Engenharia Agronômica, Universidade Estadual do Tocantins (UNITINS), Campus de Palmas, Palmas 77001-090, TO, Brazil; tiago.d@unitins.br; 9Departamento de Entomologia, Universidade Federal de Viçosa (UFV), Viçosa 36570-900, MG, Brazil; eugenio@ufv.br

**Keywords:** alternative control, *Anadenanthera colubrina*, *Curvularia lunata*, *Dalbulus maidis*, fungicide, insecticide, molecular docking, oxygen-reactive species, stunting complex, *Zea mays*

## Abstract

**Simple Summary:**

With the growing demand for sustainable and safe agricultural practices, plant compounds emerge as a solution for the alternative management of large crops. Here, we assessed the potential of using the essential oil of *Morinda citrifolia* to induce plant resistance and angico gum (*Anadenanthera colubrina*) to control phytopathogens (*Curvularia lunata*) and insect pests (*Daubulus maidis*). The chromatographic analysis identified octanoic acid (58.43%) as the main essential oil component. In resistance induction tests, plants with preventive treatment had an increase in antioxidant enzymes and chitinase. Preventive treatment with essential oil and octanoic acid (25.0 µL mL^−1^) controlled Curvularia leaf spot. Regarding the inhibition of mycelial growth, the *M. citrifolia* essential oil was potentialized with the addition of angico gum. Furthermore, molecular docking modeling revealed stable interactions between octanoic acid and the tyrosine-tRNA ligase from *Curvularia lunata*, suggesting the inactivation of protein synthesis and action specificity. The effect on corn leafhopper mortality is 76% of the population after 48 h of contact. The repellency effect in the field affected 50% of the population of adult insects, transmitters of the corn stunt disease complex. In conclusion, the study reinforces the promising use of essential oil as a biostimulant and biological control agent. It is effective as a fungicide (*Curvularia lunata*) and an insecticide (*D. maidis*) and an effective and sustainable biorational alternative.

**Abstract:**

With the growing demand for sustainable and safe agricultural practices, plant compounds emerge as a solution for biological activities. Here, we evaluated the potential of using *Morinda citrifolia* essential oil to induce plant resistance and to control phytopathogens (*Curvularia lunata*) and insect pests (*Daubulus maidis*). We conducted a chromatographic analysis to unveil the essential oil components. We also quantified the activity levels of antioxidant enzymes and chitinase for resistance induction. The antifungal action was evaluated through disease progression and the inhibition of mycelial growth in addition to in silico studies that made it possible to predict the interaction site between the fungal protein and the compounds. We assessed the toxicity and repellent actions towards the *D. maidis.* Octanoic acid (58.43%) was identified as the essential oil major compound. Preventive treatment with essential oil and octanoic acid (25.0 µL mL^−1^) increased not only the plant defense activities (i.e., the activity of the enzymes superoxide dismutase, catalase, phenol peroxidase, ascorbate peroxidase, and chitinase) but also controlled *Curvularia* leaf spot. The stable interactions between octanoic acid and tyrosine-tRNA ligase from *C. lunata* suggested protein synthesis inactivation. The essential oil inhibited 51.6% of mycelial growth, and this effect was increased to 75.9% with the addition of adjuvants (i.e., angico gum). The essential oil reduced 76% of the population of *D. maidis* adults and repelled 50% of the number of *D. maidis* after 48 h under field conditions. The repellency effect in the field reduced the population of *D. maidis* adults, transmitters of the stunting complex, by 50%. The results highlight the potential of *M. citrifolia* as a resistance activator, fungicide, insecticide, and an effective biorational alternative.

## 1. Introduction

Maize cereal (*Zea mays* L.) is the third most cultivated food in the world [1]. Brazil, with an estimated production above 110.963.7 million tons in 2024 [2], is the third-largest producer globally. Curvularia leaf spot, caused by *Curvularia lunata*, is one of the emerging diseases in the maize plant [3]. The main disease is stunting, transmitted by the leafhopper *Dalbulus maidis*, and significantly impacts productivity [4]. These occurrences are related to the increase in maize planting areas, especially during the off-season, when pathogen inoculums are maintained [5] and could be dispersed by the vector *D. maidis*. The presence and dispersion of this pathogen and vector mainly put the productivity of grains at risk. However, this production could be compromised by biotic factors, such as pests and diseases, that are increasingly resistant to synthetic compounds intensively used conventionally [6,7].

Thus, biorational strategies that are effective, economical, and environmentally sustainable have been searched for the control of pathogens and vectors. In this sense, the compounds from the secondary metabolism of plants have been reported as fungicides, insecticides, repellents, or attractants and resistance inductors in laboratory conditions [8,9,10,11,12]. Among plant species with potential use as a source of biomolecules, noni (*Morinda citrifolia* L.) plants have been shown to exert toxicity against pathogens and insects [13,14], and their compounds as elicitors have the potential of inducer resistance in plants [15]. In addition, *M. citrifolia* has been widely used in traditional medicine, is safe for mammals, and is considered an affordable source of raw material for several uses [10,16,17].

However, despite their insecticidal activities, essential oils and their compounds are biorational products that can rapidly degrade in the environment. This degradation results in limited persistence in the field, often leading to almost no residual toxic action against pests [18]. These characteristics motivate the search for new formulations or technologies to stabilize these products without a loss of biological properties. In this sense, the vegetal gum (exudates) could fulfill this role with a possible synergic effect against microorganisms by their bactericidal and fungicidal characteristics. Additionally, gums have been used as excellent vehicles for active substances, controlling the velocity of diffusion [19,20]. In this sense, the gum produced by the trunk of *Anadenanthera colubrina* (Fabaceae) [21], a plant popularly known as white angico and primarily cultivated in the Cerrado region of Brazil, consists of complex water-soluble polysaccharides (e.g., arabinose, mannose, galactose, glucopyranose, rhamnose, and hexuronic acid) [22,23]. Due to these properties, this gum has potential applications and emulsion formation with essential oils and their compounds.

Although the fungicidal potential of essential oils has been demonstrated, few studies have addressed, in an integrated way, the effect on the pathogen, its vector, and the plant. Thus, essential oils or their main compounds with pesticide potential have shown elicitor effects in plants, triggering pathogen-related reactions similar to those directly mediated by pathogens (e.g., chitinase activities) or indirectly activating latent defense mechanisms [12,24,25]. However, phytotoxic effects on plants have also been reported [26,27], including *M. citrifolia* essential oil [9,28]. Therefore, the unintended effects on plants must be discussed, such as the phytotoxicity, elicitor properties, or action mode of these products. Thus, we aimed here to evaluate the fungistatic activity of *M. citrifolia* essential oil (in a mixture with angico gum) against *C. lunata*. We further assessed the toxicity and repellence of this biorational product against *D. maidis*, the vector of mollicutes. Additionally, we measured the potential phytotoxic effects and the biochemical responses on the maize plants elicited by the *M. citrifolia* essential oil. By applying bioinformatics tools, we were able to predict potential interactions of the *M. citrifolia* essential oil major component (i.e., octanoic acid) and the pathogen-related tyrosine-tRNA ligase.

## 2. Materials and Methods

### 2.1. Obtaining Morinda citrifolia Essential Oil

*Morinda citrifolia* L. plants were identified in the Gurupi region, Tocantins, Brazil, for material collection. The essential oil was extracted from ripe fruits, previously washed and macerated for insertion into a volumetric flask by steam hydrodistillation in a modified Clevenger-type device and stored at four °C until analysis [29]. The major compound, octanoic acid, was purchased from an industry specialized in producing this isolated compound (Sigma–Aldrich^®^, São Paulo/SP, Brazil).

### 2.2. Gas Chromatography–Mass Spectrometry (CG-MS)

Identifying essential oil constituents was conducted using a Shimadzu QP2020 chromatograph equipped with a QP2010 Plus selective mass detector. This was operated under an RTX-5MS fused silica capillary column (30 m × 0.25 mm × 0.25 μm), with a programmed temperature range of 50 to 280 °C (increasing at a rate of 3 °C min^−1^). The carrier gas used was helium, and a splitless injection method was used, with an injected volume of 1 μL of a 1:1000 solution in hexane. The mass spectrometer, configured with an impact energy of 70 eV, had an ion source and interface temperature of 280 °C. The resulting spectra were compared with those from the National Institute of Standards and Technology (NIST) and Wiley 229 databases. The retention rate of each constituent was compared with the standards provided by Adams [30], and the values were expressed as percentages.

### 2.3. Purification of Anadenanthera Colubrina Gum

The tree of *Anadenanthera colubrina* was identified in the Gurupi region, Tocantins, Brazil, to collect vegetal gum (exudates). To purify the angico gum, 5.0 g of the crushed material was added to 100 mL of water, adding 0.5 g of NaCl and hydrogen peroxide. The system was placed under orbital agitation for 12 h at room temperature. Then, the material was filtered using qualitative filter paper with a vacuum pump model LT 65 (LIMATEC). After the first filtration stage, 200 mL of 99% ethyl alcohol was added, and the system was maintained at a temperature of 4 °C for precipitation. After the precipitation step, the material was centrifuged at 3000 rpm for 5 min in an Excelsa II MOD centrifuge, 206 BL (FANEM), and the precipitate was stored to repeat the steps thrice [31].

### 2.4. Phytotoxicity of M. citrifolia and Anadenanthera colubrina in Corn Leaf

Maize seeds (cultivar 30A37-PM) were grown in 8 L pots with equal parts soil and commercial substrate. The experiment was carried out in a greenhouse. Thirty days after sowing, manual sprayers were used to apply 5 mL of *M. citrifolia* essential oil solution (3.0, 9.0, 15.0, 20.0, and 30.0 µL mL^−1^) and angico gum (*Anadenanthera colubrina*) at concentrations of (0.3, 0.9, 1.5, 2.0, and 3.0 µg mL^−1^). After 24 h of application, the scale of phytotoxicity was determined according to Brum et al. [32] by assigning values referring to the percentage of damaged leaf area, which were as follows: 0 = absence of phytotoxicity; 1–25% = mild leaf necrosis or mild chlorosis of the plant; 26–50% = moderate leaf necrosis or moderate chlorosis of the plant; 51–75% = high leaf necrosis or high chlorosis of the plant; and 76–100% = wilt and dryness of the plant.

### 2.5. Determination of Biochemical Effects on Corn Plants

Maize seeds (cultivar 30A37-PM) were grown in 8 L pots in a greenhouse with equal parts soil and commercial substrate. Thirty days after sowing, manual sprayers were used to apply 5 mL of essential oil solution at concentrations of (6.3, 12.5, 25.0, 50.0, and 75.0 µL mL^−1^). In parallel, other plants were treated with octanoic acid at the same concentrations. Two hours after application, the parcel of plants (containing pathogen) was inoculated with 1 × 10^6^ conidia mL^−1^ of the fungus *C. lunata*. All treatments were performed in triplicate. Ten days post-inoculation, enzymatic activities were monitored by obtaining 200 mg of fresh plant tissue macerated in liquid nitrogen containing 20% polyvinylpyrrolidone and homogenized with potassium phosphate buffer (100 mM; 7.0 pH), added EDTA (1 mM), and ascorbic acid (1 mM). The mixture was centrifuged for 25 min (10,000 rpm) at a temperature of 4 °C, and the protein extract was stored at −20 °C for later analysis.

Superoxide dismutase (SOD) quantification was carried out according to the methodology of Beauchamp and Fridovich [33]. Potassium phosphate buffer (50 mM; 7.8 pH) was added to a solution containing EDTA, L-methionine (70 mM), nitroblue tetrazolium (NBT) (1 mM), and riboflavin (2 mM). After adding the protein extract, the mixture was placed in a lighted chamber. Absorbance was measured at 540 nm (BioSpectro, Shangai, China, model SP-2000UV) [34]. Activity was expressed in units per gram per mass per extract per minute (U g^−1^ E^−1^ min^−1^).

The activities of the enzymes catalase (CAT), phenol peroxidase (POX), and ascorbate peroxidase (APX) were measured by the decrease in absorbance monitored for 3 min with readings every 15 s. The CAT analysis method was modified from the [35] test. Potassium phosphate buffer (200 mM; 7.0 pH) was mixed with hydrogen peroxide, and the reaction started with protein extract. Absorbance was read at 240 nm, and activity was expressed as micromoles of hydrogen peroxide per gram per extract per minute μmol H_2_O_2_ (g^−1^ E^−1^ min^−1^). POX activity was analyzed based on the oxidation of phenolic compounds, as described by Amako et al. [36], which promotes a decrease in absorbance at 470 nm. Sodium acetate buffer (0.2 M; 5.0 pH), added guaiacol (0.2 M), hydrogen peroxide (0.38 M), and protein extract were used. Activity was expressed in micromoles of hydrogen peroxide per gram per extract per minute (µmol H_2_O_2_ g^−1^ E^−1^ min^−1^). The APX enzyme activity determined by the ascorbate oxidation rate (ASA) was quantified using the method described by Asada [37]. The solution containing potassium phosphate buffer (200 mM; 7.0 pH), ascorbic acid (10 mM), and hydrogen peroxide (2 mM) was prepared, and the reaction started with protein extract. Activity was monitored at 290 nm and expressed as micromoles of ascorbate per gram of extract per minute (μmol ASA g^−1^ E^−1^ min^−1^).

For the chitinase enzyme reaction (QUIT), as described by Wirth and Wolf [38], a reaction medium was prepared containing sodium acetate buffer (0.1 M; 5.0 pH), “CM-chitin-RBV^®^” (2.0 mg mL^−1^), and protein extract. The medium was incubated (40 °C; 20 min), and the reaction stopped with the addition of HCl (1 M), cooling, and centrifugation (14,000 rpm; 5 min). The absorbance of the supernatant was monitored at 550 nm, and activity was expressed in absorbance units per minute (U min^−1^).

### 2.6. Molecular Docking

#### 2.6.1. Ligands

Molecular docking was performed using the major compounds of *M. citrifolia*. The 3D structures of the compounds in their neutral forms were constructed using Marvin Sketch 18.10, ChemAxon “http://www.chemaxon.com (accessed on 2 March 2023)”.

#### 2.6.2. Target Modeling

Amino acid sequences of tyrosine-tRNA Ligase from *Curvularia lunata* (*Cochliobolus sativus* sequence was used because of its high similarity—M2TAQ3) were obtained from the UniProt “http://uniprot.org (accessed on 3 March 2023)” database. The 3D structures of both proteins were constructed using a homology modeling approach in the Swiss Model Workspace “https://swissmodel.expasy.org/ (accessed on 3 March 2023)” after selecting their respective templates using the BLASTp tool. The templates were downloaded from The Protein Databank “https://www.rcsb.org/ (accessed on 3 March 2023)”, considering the quality parameter experimental method, resolution, R-value, and complexation with a ligand. Protein structure crashes and amino acid positioning at the active site were checked using the Swiss model [39]. The generated models were validated with inspection of the Ramachandran plots [40,41], which analyze the distribution of the torsion angles of the backbone ɸ and ψ responsible for the stereochemical quality of a protein as well as the QMEAN factor [42].

#### 2.6.3. Molecular Docking Calculations

Targets and ligands were prepared for molecular docking using Autodock Tools 1.5.7 [43] following de Souza Moura et al. [44]. In AutoDock Vina (Trott and Olson, 2010), nine docking positions were generated for the ligands interacting with the targets, returning the affinity energy values (kcal/mol). The docking position results were analyzed using PyMOL 2.0 and Discovery Studio 4.5 [45] to select the best position for each ligand inside the protein target using the parameters.

#### 2.6.4. Molecular Dynamic Simulations

Molecular dynamic simulations were performed using the MDWeb server [46]. Molecular docking PDB files were used to prepare the simulation base. First, the structure was prepared using the Gromacs complete molecular dynamic (MD) setup with the AMBER-99SB × force field, as it satisfactorily describes the molecular behavior of proteins. Then, the molecular dynamic simulation process was carried out according to the following steps: cleaning the structure, fixing the side chains of the complex, adding the solvent box, minimizing energy, and balancing the system to receive the minimized structure as an outlet [47]. These simulations were performed at constant number, constant volume, and temperature (NVT) [48,49]. In the equilibrium stage, the systems were simulated for 2.5 ps at a temperature of 27 °C and constant pressure. After generation of the protein-ligand complex, the water molecules and ions were removed to reduce the size of the system, and the dry trajectory was recovered to trace the root-mean-square deviation (RMSD).

### 2.7. Curvularia Leaf Spot Preventative Control

Maize seeds (cultivar 30A37-PM) were grown in 8 L pots with equal parts soil and commercial substrate. The experiment was carried out in a greenhouse. Thirty days after sowing, manual sprayers were used to apply 5 mL of essential oil solution at (6.3, 12.5, 25.0, 50.0, and 75.0 µL mL^−1^) concentrations. In parallel, other plants were treated with octanoic acid at the same concentrations. Two hours after application, the plants were inoculated with 1 × 10^6^ conidia mL^−1^ of the fungus *Curvularia lunata*, which was acquired from the library of the Department of Phytopathology at the Federal University of Tocantins (Gurupi, Tocantins, Brazil). All treatments were performed in triplicate. Control plants were inoculated under the same conditions with sterilized distilled water. Disease severity was assessed every two days post-inoculation, with a total monitoring period of 10 days [50].

### 2.8. Inhibition of Mycelial Growth of Curvularia lunatalll

The bioassay was conducted using 70 mm diameter Petri dishes, applying four distinct treatments: (I) 15.0 µL mL^−1^
*M. citrifolia* essential oil, diluted in sterilized water with 10.0 µL mL^−1^ of Tween 20; (II) 1.5 µg mL^−1^ angico gum, diluted in distilled water; (III) 1.5 µg mL^−1^ angico gum combined with 15.0 µL mL^−1^ essential oil; and (IV) sterile distilled water with 10.0 µL mL^−1^ Tween 20 as the control. A completely randomized experimental design was used in a factorial scheme, with three replications and five evaluations (2, 4, 6, 8, and 10 days of incubation).

Subsequently, 200 µL of each solution (i.e., essential oil, angico gum, and essential oil + angico gum) were distributed on the surface of the commercial PDA culture medium (Potato-Dextrose-Agar) in separate Petri dishes with a Drigalski loop. Then, a disc (4 mm) containing the mycelium of the fungus *C. lunata* was deposited in the center of each Petri dish. The plates were sealed, identified, and kept in an incubation chamber at 25 °C for ten days.

The diameter of the colonies was measured on the vertical and horizontal axes at regular intervals of 48 h. The diameter of the colonies was measured on the vertical and horizontal axes at regular intervals of 48 h. From the values obtained for the fungus’s average diameter, the mycelial growth rate index (IVCM) was calculated, adopting the formula described by Maia et al. [51]. The measurements of average diameters measured in each evaluation, reduced from the control value, were adopted as ideal for the percentage of inhibition of mycelial growth.

### 2.9. Essential Oil and Angico Gum Effect on Dalbulus maidis

#### 2.9.1. Toxicity

The *D. maidis* collected from crops in the region were kept in cages with anti-aphid screens in a greenhouse at room temperature. Five healthy maize plants, with four expanded leaves, were placed into the cage, allowing for the insects to lay eggs. Subsequently, 10 adult insects were transferred to the plants for laying eggs during a period of four to six days; after this period, the insects were removed, and the newly hatched larvae were kept until they became adults. These adult insects were collected with an insect suction pipe and placed in gerbox boxes covered with organza fabric. Each box contained a cotton pad moistened at the ends of the maize leaf, which was available for feeding in a transverse direction to avoid dehydration.

The insects were deposited in the environments, and 750 µL of each treatment was sprayed. The following concentrations (2.5, 5.0, 10.0, 15.0, 20.0, 30.0 µL mL^−1^) of *Morinda citrifolia* essential oil were used. Emulsions were prepared with the following concentrations of essential oil (5.0, 10.0, 15.0, 20.0, and 30.0 µL mL^−1^) and fixed amounts of angico gum previously determined (1.5 µg mL^−1^) added with tween 20 (10.0 µL mL^−1^). As a control, identical amounts of sterilized distilled water were sprayed. The experiment was carried out in triplicate, and survival was assessed for 72 h at a 24 h interval.

#### 2.9.2. Field Repellency

The experiment was carried out in a field in the tropical region in the municipality of Gurupi, Tocantins, Brazil, with maize crops in the 2023/24 agricultural harvest. In a traditional commercial area considered to have high disease pressure, private property is located in a rural area (48°53′34″ W; 11°44′29″ S).

The experiment was carried out in an area of approximately 260 m^2^. Soil preparation was carried out conventionally, using a plow harrow. Base fertilization (N-P-K) was carried out depending on the soil analysis results—urea coverage 20 days after planting. The cultivar B2433PWU was sown with a spacing of 0.9 m between rows. The plant population was 65 thousand ha^−1^. The experiment was implemented using a randomized block design with four replications. Each plot was ten m^2^ in size. With experiments previously carried out under controlled conditions, the following treatments were selected for field evaluation: *M. citrifolia* essential oil at a concentration of 15.0 µL mL^−1^; an emulsion prepared with 1.5 µg mL^−1^ angico gum and *M. citrifolia* essential oil (15 µL mL^−1^) added with tween 20 (10.0 µL mL^−1^); the insecticide Lannate ^®^BR (CORTEVA AGRISCIENCE) (600 µL L^−1^); and sterile distilled water only sprayed as a control. Two plants per block were evaluated.

### 2.10. Statistical Analysis

Area under the disease progression curve (AUDPC) was calculated according to Shaner and Finney [52]. The results of AUDPC, inhibition of mycelial growth, and induction of enzymatic activities and insect mortality were subjected to regression analysis using curve fitting procedures in SigmaPlot 12.5 (Systat Software Inc., San Jose, CA, USA). Reference values are shown in the Appendix A. The models were selected using the parsimony criterion, checking the assumptions of normality and homogeneity of variance. For the mycelial growth rate, mycelial diameter growth, and repellency tests, an analysis of variance (ANOVA) was performed.

## 3. Results

### 3.1. Gas Chromatographic Analysis (GC-MS) of Morinda citrifolia Essential Oil

The essential oil’s chromatographic analysis revealed a higher composition of octanoic acid (58.43%), hexanoic acid (9.46%), pent-4-enyl hexanoate (8.17%) at methyl octanoate (7.16), and other compounds with a mass percentage above 1% (Table 1). A total of 25 compounds were identified in the analyzed sample (Appendix A).

### 3.2. Biochemical Effects

Our result showed that octanoic acid is more efficient at activating the SOD in maize plants than the *M. citrifolia* essential oil; the SOD activation in the latter decreases with increasing concentrations. In contrast, the octanoic acid in intermediary concentrations significantly increased the enzymatic activity (97.3 U g^−1^ E ^−1^min ^−1^); however, there was a loss of this property in concentrations > 5.0 µL mL^−1^ (Figure 1a). This SOD activation, when applied to octanoic acid, was similar to treatment with *C. lunata* only, evidencing the efficiency of this compound as a preventive measure.

Similarly, the CAT enzyme in maize plants was increased with the octanoic acid alone or in combination with *C. lunata* fungi. When applied purely in low concentrations, the CAT activity is 48.6 µmol H_2_O_2_ g^−1^ E^−1^ min^−1^, three times greater than the levels found in healthy plants or the plants *M. citrifolia* essential oils (Figure 1b) or *C. lunata*. With increased concentration, the CAT activity decreases independently of treatment. Contrary to what was observed, the ascorbate peroxidase enzyme APX negatively affected the octanoic acid; higher levels were observed in plants infected with *C. lunata* and treated with *M. citrifolia* essential oil in a dependent concentration. Also, it was observed that the enzyme activity was highest for plants infected by the fungus (468.7 µmol AsA g^−1^ E^−1^ min^−1^) (Figure 1c). The phenol peroxidase (POX) was also increased in intermediary concentrations of octanoic acid or *M. citrifolia* essential oil; this increase was of 0.44 µmol H_2_O_2_ g^−1^ E^−1^ min^−1^ in healthy plant to 1.3 and 1.2 µmol H_2_O_2_ g^−1^ E^−1^ min^−1^ for the treatments with essential oil and octanoic acid (2.5 µL mL^−1^), respectively (Figure 1d). Peroxidase (POX) activity was determined in relation to the presence of *C. lunata*, showing peaks of increased activity at intermediate concentrations independent of the treatments involving *C. lunata*. However, in the absence of *C. lunata*, the variation in POX activity was higher for octanoic acid compared to *M. citrifolia*, showing a negative correlation to octanoic acid and a positive correlation to pure essential oil (Figure 1d).

Similarly, the QUIT activity was favored by the octanoic acid, which induced a greater amount of the enzyme compared to the *M. citrifolia* essential oil; however, this increase was only in intermediary concentration (Figure 1e). The QUIT production with the treatments mentioned was less in healthy plants and plants infected with the pathogen only. All statistical results of Figure 1 stay represented in Appendix A.

### 3.3. Phytotoxicity of M. citrifolia and Anadenanthera colubrina in Corn Leaf

The percentage of damaged leaf area in plants treated with different concentrations of *M. citrifolia* essential oil and angico (*A. colubrina*) gum is shown in Figure 2. In addition, no symptoms of phytotoxicity (chlorosis or necrosis) were observed in the plants treated with the lethal doses of the essential oil and angico gum estimated for *D. maidis*.

### 3.4. Molecular Docking

#### 3.4.1. Interactions of *Morinda citrifolia* Components and Fungal Tyrosine-tRNA Ligase

Ramachandran-favored values and the QMEAN factor of the *Curvularia lunata* (with *Cochliobolus sativus* identity to 52.89%) and tyrosine-tRNA ligase protein model were 95.04% and −0.73, respectively (Appendix A). Tyrosine-tRNA ligase exhibited higher energy affinity when compared with octanoic acid (−5.0 Kcal mol^−1^) (Appendix A). Besides hexanoic acid (−4.2 Kcal mol^−1^), there were 4-pentenyl ester [pent-4-enyl hexanoate] (−4.6 Kcal mol^−1^), octanoic acid, methyl ester [methyl octanoate] (−4.4 Kcal mol^−1^), hexanoic acid, methyl ester [methyl hexanoate] (−4.1 Kcal mol^−1^), and isobutyl 3-methylbut-3-enyl carbonate [3-methylbut-3-enyl 2-methyl propyl carbonate] (−4.6 Kcal mol^−1^). The complex of *C. lunata* formed between the octanoic acid and the tyrosine-tRNA ligase showed a close link to the amino acids of the active site (Figure 3a). This complex comprised alkyl interactions (VAL97, PRO98, and VAL258), van der Waals interactions (PHE226, GLY228, ASP230, ASN255, PRO256, and MET257), conventional hydrogen bonds (TRP83 and GLN231), and carbon–hydrogen bonds (GLY227) between the target’s active site and the ligand (Figure 3b).

#### 3.4.2. Molecular Dynamic Simulation

In the dynamic molecular simulation, the spatial RMSD was calculated from the average position of each amino acid residue of the complexes formed by the OA ligand and the protein from the tyrosine-tRNA ligases of *Curvularia lunata* to confirm structural stabilization (Appendix A). The highest RMSD values were below 1.0 Å for residues that underwent significant changes only in the regions corresponding to the loops. In contrast, for the residues in the active-site region, the RMSD value was lower, revealing the stability of these areas.

### 3.5. Curvularia Leaf Spot Control

The area under the disease progress curve (AUDPC) was dependent on the concentration of the treatment applied. When treated with essential oil in lower concentrations alone, the (AUDPC) increases, but in high concentrations, the damaged area of the leaves is reduced significantly (Figure 4). When treated with octanoic acid, this phenomenon (increase) was not observed, and the AUDPC was reduced according to the concentration applied, the reduction being greater than the *M. citrifolia* essential oil.

### 3.6. Effect of Morinda citrifolia Essential Oil and Gum Anadenanthera colubrina on the Development of Curvularia lunata

Our results show the growth of mycelium *C. lunata* when exposed to *M. citrifolia* essential oil alone or in combination with gum from *A. columbrina*. Both treatments significantly reduced the growth of the fungus over time (Table 2). After the tenth day of incubation, the mycelium growth of *C. lunata* was 37.6 ± 1.1 mm on the treatment of *M. citrifolia* essential oil, being reduced significantly (*t* = 24.01; *df* = 4; *p* < 0.001) (Figure 5a); in the mixes of angico gum and *M. citrifolia*, the growth also was reduced significantly (*t* = 5.84; *df* = 4; *p* = 0.004) (Figure 5b).

### 3.7. Preventive Measure

#### 3.7.1. Toxicity in Field to Vector *Dalbulus maidis*

The leafhopper *D. maidis* the main vector of *C. lunata* also was susceptible mainly to one combination of gum + *M. citrifolia* essential oil. This mortality was time dependent and treatment dependent, without difference at 24 h (t = 2, *df* = 4, *p* = 0.116) and being higher after 48 h (t = 4, *df* = 4, *p* = 0.016) and 72 h (t = 5, *df* = 4, *p* = 0.007) (Figure 6).

#### 3.7.2. Repellence in Field

Our results indicate that repellence to *D. maidis* is significantly influenced by the applied insecticide solution, the vegetative stage of the plant, and the interaction between these factors. Additionally, we could record the effects of time and its interaction with the plant vegetative stage (Appendix A). The number of *D. maidis* found in maize plants was less in the early stage (V3) without difference between the treatments (Figure 7a); however, the adult number of *D. maidis* in the V7 stage is higher, with an effect of the treatment and the time in the presence of insect (Figure 7b).

## 4. Discussion

Research into the use of *Morinda citrifolia* has little exploration in agriculture. The investigated compounds have not been reported previously for potential antifungal action against *Curvularia lunata*, as an insecticide (*Dalbulus maidis*), and as a resistance inducer in maize. The chromatographic analysis indicated that the components with the most significant levels are carboxylic acids. Studies conducted with noni oil obtained from Brazil and Malaysia revealed significant content disparities, up to 20% variations. However, it is worth highlighting that those fatty acids remain these oils’ predominant compounds [28,53]. Thus, targeting compounds of interest with proven beneficial effects optimizes the production of the isolated compound or potentialized essential oil.

The chemical composition of natural compounds is the main factor determining their stability, revealed by a thermal analysis and characterizations [54,55]. Using an isolated compound as an adjuvant showed promise, protecting against degradation. Understanding natural compounds’ chemical composition and thermal behavior is essential to assess their stability, potential application in varied climatic conditions, and specific functions. The polymeric characteristics and thermal analyses confirm similar structures between polymers [23,56], allowing for the resin to act as an ideal compound in forming an emulsion with an essential oil. The complexed product maintains the biological activities of interest and increases the persistence period in the environment.

In addition to the biological action, it is necessary to confirm the low phytotoxic effect of alternative controls for foliar application. The phytotoxicity described by Werrie, Durenne, Delaplace, and Fauconnier [26], such as adverse impacts caused by the biostimulation of compounds, may be related to cellular dysfunctions and must be carefully evaluated [57]. The effects of *M. citrifolia* essential oil and angico gum at concentrations below 3.0% did not cause levels of chlorosis or necrosis capable of affecting photosynthetic mechanisms. Abd-Elgawad [58] compared the dose-dependent phytotoxicity effects of essential oils rich in terpenes and maintained the same pattern in different crops, confirming the need for preliminary analyses to determine ideal concentrations. The action mechanisms depend on pathogens’ biochemical response in plant biosynthesis pathways. They interfere with protein synthesis and respiration, degrading the membranes’ integrity and cytoplasmic content, inhibiting electron transport, and inactivating mitochondrial activity [59].

Compounds from *M. citrifolia* activate plants’ latent defense mechanisms and the antioxidant system. The analyses of enzymes and their mode of action in healthy plants and those infected with *Curvularia lunata* provide insights into the pathogen–host interaction. Superoxide dismutase (SOD) is activated strongly in the presence of octanoic acid with or without the pathogen in intermediate concentrations tested here, being equal to cause expression by the pathogen alone, showing this compound as a potential resistance activator to the fungal infection of *C. lunata* in maize plants.

Catalase, unlike APX or POX, degrades H_2_O_2_ without requiring a reducing agent (such as ascorbate or glutathione), and its activity was enhanced by octanoic acid in triggering pre-fungal infection resistance. The same plant compounds described by [15] also induce the enzymes CAT, SOD, and APX, improving the preventive control of pathogens in *Cucumis melo* L. Unlike CAT, APX and POX depend on cofactors for H_2_O_2_ reduction. Studies with essential oils have shown that *Cymbopogon* sp. and *Cymbopogon citratus* increase resistance against *Pseudomonas syringae* [60]. It is known that the antioxidant responses of SOD, CAT, APX, and POX in *Zea mays* and other species vary according to the level of stress (biotic and abiotic) [61,62]. Phenol peroxidase (POX) demonstrated the lowest activity, suggesting that the action depends on the availability of cofactors. Despite these low values, the fatty acid raised the enzyme level as a preventive treatment.

The activity of chitinase, crucial in the degradation of the pathogen’s cell wall [63], was increased with preventive treatment with octanoic acid, enhancing the enzymatic response. Likewise, bacterial isolates against *Fusarium verticillioides* in maize, described by Figueroa-López et al. [64], increased chitinase and demonstrated the relationship between reduced incidence and antagonistic activity with the pathogen.

The specificity activity and inhibitory effect of fungi and bacteria initiated by computational analyses of active-site docking indicate important parameters for predicting mechanisms of action [65]. The fungal enzyme selected for modeling plays an important role in protein biosynthesis. Therefore, compounds that compete with this enzyme’s activation site are potential antimicrobials [66]. With in silico approaches, the interactions between tyrosine-tRNA ligase and the main compounds of the essential oil showed greater stability with octanoic acid, as reported [10]. Energy levels close to those found reinforce the results obtained with the efficiency of octanoic acid in similar action against fungi that cause important diseases [9]. The effect of the ligand (octanoic acid) and target receptor (Tyrosine-tRNA ligase) is influenced by the types of bonds and their intensity. Those involved in the present study are conventional hydrogen bonds, alkyl interactions, van der Waals interactions, and carbon–hydrogen bonds, which characterize high affinity. These affinities corroborate the suggestion that stable interactions compromise the synthesis of fungal enzymes.

Studies on the containment of foliar diseases that threaten large crops, such as emerging diseases, emphasize the importance of alternative controls [67]. The assessment of the disease’s progress made it possible to prove the efficiency of *M. citrifolia* and octanoic acid in defending against the phytopathogen *Curvularia lunata* as well as studies with other essential oils that demonstrate the susceptibility of the fungus to fatty acids [23,68]. Direct action on fungal growth accompanied by mycelial growth inhibition demonstrated the enhancement of the essential oil’s fungistatic capacity with the incorporation of angico gum. In addition to reducing the speed of mycelial growth, this reinforces the viability of using fatty acids and biopolymers in new formulations for sustainable management.

The toxicity of fatty acids against fungi has also been reported with fungistatic and fungicidal activity against *Stagonosporopsis cucurbitacearum*, *Aspergillus niger*, *Alernaria solani*, *Fusarium oxysporum*, *Botrytis cinerea*, and *Saccharomyces cerevisiae*, reinforcing the hypothesis of the use of fatty acids as alternative fungicides acting on cell membrane degradation [9]. Biorational compounds stand out for their lower environmental risk [69] and specific action, which results in fewer possibilities of acquiring resistance.

The association of resistance induction, antifungal activity, and action against insect pests shows the capacity and complexity of mechanisms involved in using natural compounds as broad-spectrum bioproducts. The corn leafhopper is considered a primary pest in the Neotropical region, and it causes direct and indirect damage by transmitting diseases caused by mollicutes and viruses [70]. The toxicity of *M. citrifolia* essential oil was verified over 72 h. Therefore, lethality, dependent on dose and time, suggests systemic action after contact with the compound. In the face of higher concentrations, a population containment capacity of 65% emphasizes the investigated activity, even when working with a volatile compound without adjuvants. The potential pest control with reported plant compounds proves the susceptibility of insect populations of the same order and family to the rosemary essential oil, *Eucalyptus globulus* L., and sesquiochemicals of *Tagetes erecta* and *Flemingia macrophylla*, both by repellency and lethality, ensuring their use [69,71,72]. Thus, our results demonstrated that despite few studies on the mechanisms of action, the insecticidal and repellent activities of compounds from *M. citrifolia* and angico demonstrated the potential additive effect of the oil associated with the gum.

Systemic stunt infections are caused by bacteria from the mollicute class, which colonize the phloem. The main consequences are the impairment of chlorophyll levels and the limitation of nutrient production and absorption pathways [73]. There is a need for instant action due to the high population numbers and displacement that spread disease in the early stages of maize development. The essential oil action, which began within the first six hours of contact, allows us to suggest consecutive applications to optimize efficiency and achieve high levels of population control. *M. citrifolia* essential oil and angico resin’s combined action potentiated the response in the first six hours after application. It reached control levels above 60% for all concentrations that with essential oil alone, did not exceed 10% mortality. These results suggest that the adjuvant circumvents the volatility of natural compounds.

The toxic effect on insects under controlled conditions was also expanded at the field level to verify the repellency action at different stages, where the crop is more susceptible to insect attacks. As reported in another study [10], the initial stages of development have the highest incidence of oviposition and, consequently, of nymphs compared to the number of adult insects constantly moving to guarantee survival. In the first evaluation carried out in stage V3, both *M. citrifolia* essential oil and the emulsion showed a reduction among the treatments within 48 h. However, the emulsion reduced the number of insects by half and contained the population increase, which can be justified by the more notable persistence with the addition of the biopolymer. The results corroborate several studies using essential oils in pest control [74,75,76]. The repellency patterns of adults in stage V7, in which the number of insects was much higher than in the first evaluation, better demonstrated the efficiency of the essential oil and the emulsion and maintained a level of control above expectations.

Alternative control is encouraged because it can minimize the effect of pesticides, with proven efficiency and biodegradation and low toxicological risks. These aspects lead to selectivity studies of beneficial organisms, such as pollinators, predatory species of natural enemies, soil microorganisms, and aquatic bioindicators [13,77,78]. Once the action against the phytopathogenic fungus (*C. lunata*) and the insect pest (*D. maidis*) has been proven, it is necessary to evaluate the selectivity of *Morinda citrifolia* essential oil against natural enemies. According to other investigations [10], the low toxicity of *M. citrifolia* essential oil was verified against non-target organisms, *Coleomegilla maculata* and *Eriopis conexa*, crucial biological control agents, as well as *Trichoderma asperellum*, an important antagonist agent used against phytopathogenic fungi, such as *Fusarium graminearum*, *F. verticillioides*, and *Rhizoctonia solani* [79,80]. *T. asperellum* is also identified by [81] as an insect pest inhibitor of maize (*Ostrinia furnacalis*).

Therefore, the results demonstrate that compounds from *Morinda citrifolia* have activities that characterize them as potential biorational controls for managing Curvularia leaf spot and the vector transmitting the rickets complex (*D. maidis*). Furthermore, they did not present antagonistic effects on non-target organisms or stimulate plant defense mechanisms. Thus, our results demonstrated that noni essential oil and its emulsion with angico gum had great potential for use in the biorational control of phytopathogens and insect vectors of diseases in maize with low environmental impact.

## 5. Conclusions

The *Morinda citrifolia* secondary metabolism constituents effectively controlled Curvularia spot without triggering toxic effects for maize plants. With in silico approaches, the interaction of the fungal protein with the potential target of octanoic acid makes it possible to predict the relationship with disease control. It also presents toxicity to the insect pest (*Dalbulus maidis*) of maize crops and the vector of the stunting complex. In addition to the repellent action in the field in the first six hours after contact and maintenance for up to 48 h, angico gum demonstrated significant insecticidal effect enhancement and fungal growth inhibition when incorporated into the emulsion with essential oil. Both octanoic acid and essential oil act as inducers of resistance and biochemical activities in healthy plants and plants treated preventively against the inoculation of the fungus. Thus, this confirms the hypothesis that they are promising activators of the antioxidant system in degrading reactive oxygen species and activating defense mechanisms with antifungal and insecticidal action. These results highlight the feasibility of biorational control with the potential use of the compounds as components of an alternative natural fungicide and insecticide in addition to activating plant resistance.

## Figures and Tables

**Figure 1 biology-13-00479-f001:**
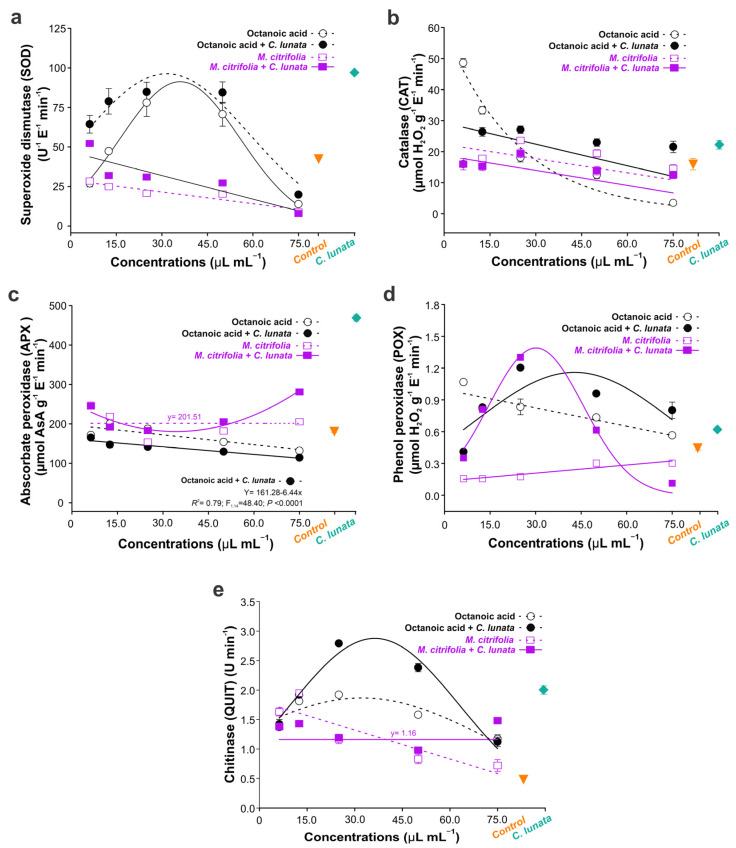
The enzymatic activity of superoxide dismutase (**a**), catalase (**b**), ascorbate peroxidase (**c**), phenol peroxidase (**d**), and chitinase (**e**), as a function of treatments with noni (*Morinda citrifolia*) essential oil (EO) and acid octanoic acid (octanoic acid) applied to maize plants. Samples were inoculated only with the pathogen *Curvularia lunata* (Pathogen) and that received preventive treatments with noni (*M. Citrifolia* + Pathogen) and octanoic acid (Octanoic Acid + Pathogen). The symbol shows the mean (±SD) of three replicates.

**Figure 2 biology-13-00479-f002:**
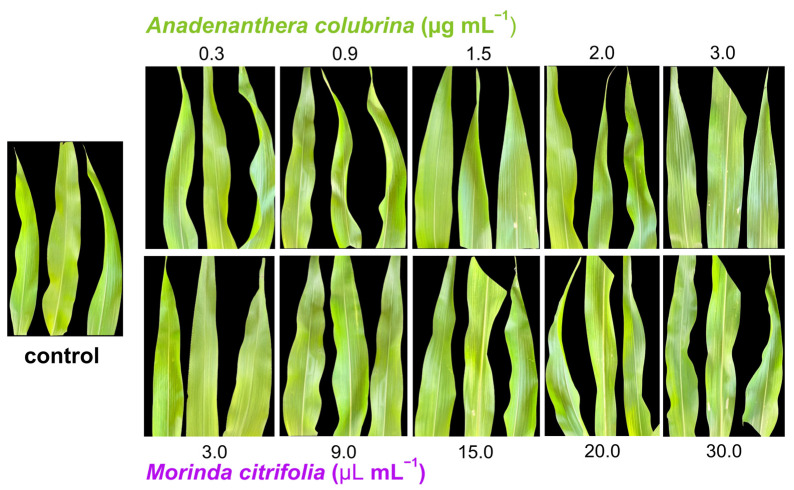
Phytotoxicity symptoms in maize plants when treated with several concentrations of *Anadenanthera colubrina* and *Morinda citrifolia* essential oil.

**Figure 3 biology-13-00479-f003:**
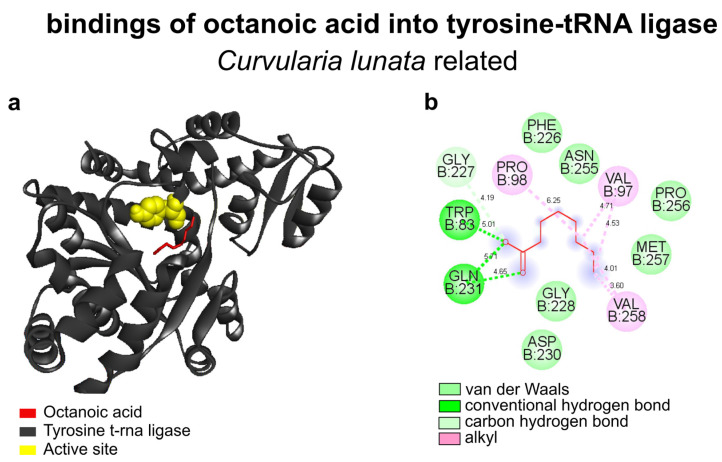
Octanoic acid (red) complexed with tyrosine-tRNA ligase (**a**) and 2D maps of molecular interactions with amino acids in related target active sites (yellow) of *C. lunata* (**b**).

**Figure 4 biology-13-00479-f004:**
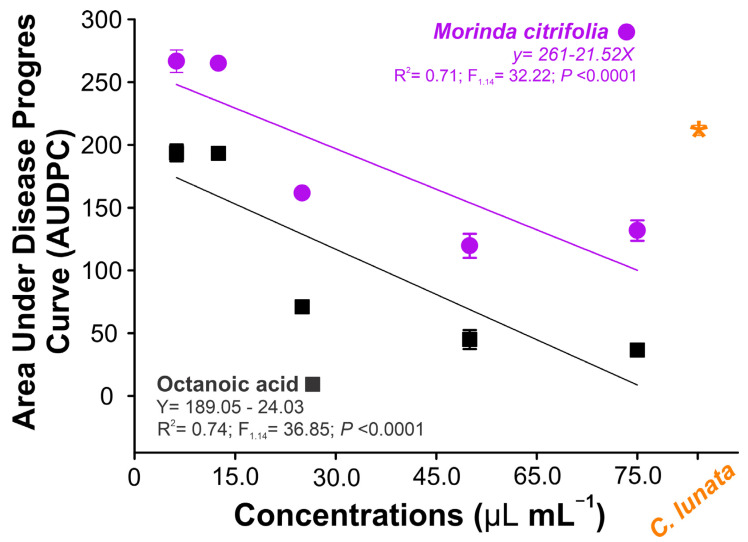
The area under the disease progress curve (AUDPC) in leaf maize plants when treated with noni essential oil (*Morinda citrifolia*) and ethanoic acid in different concentrations. The symbol shows the mean (±SD) of three replicates.

**Figure 5 biology-13-00479-f005:**
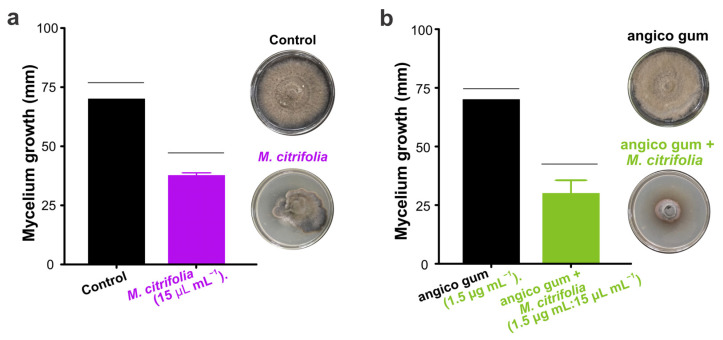
The diameter (±SE) mycelial of *Curvularia lunata* after ten days subjected to treatments with *Morinda citrifolia* essential oil (15.0 µL mL^−1^) (**a**) and a mixture of *Anadenanthera colubrina* angico gum (1.5 µg mL^−1^) and *M. citrifolia* (**b**). The bars covered with an individual horizontal line have statistical differences via *t*-test (*p* < 0.05).

**Figure 6 biology-13-00479-f006:**
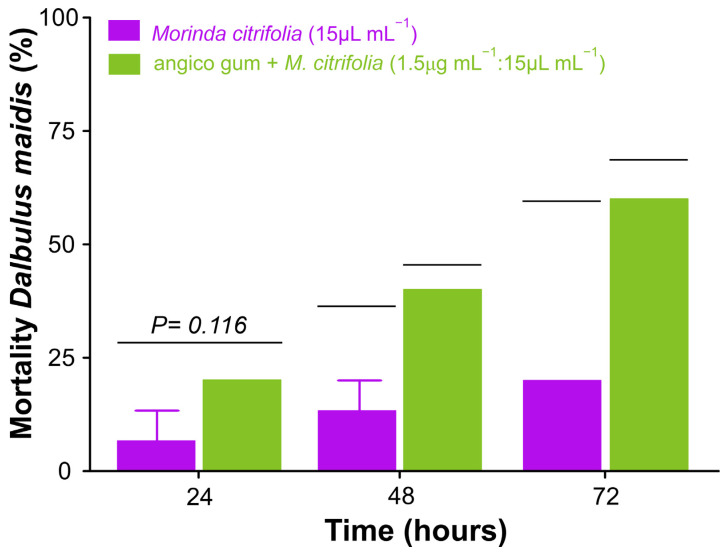
The mean (±SE) mortality of corn leafhopper, *D. maidis,* exposed to *Morinda citrifolia* essential oil and a mixture of angico gum and *M. citrifolia* after 24, 48, and 72 h. The bars covered with an individual horizontal line have statistical differences via *t*-test (*p* < 0.05).

**Figure 7 biology-13-00479-f007:**
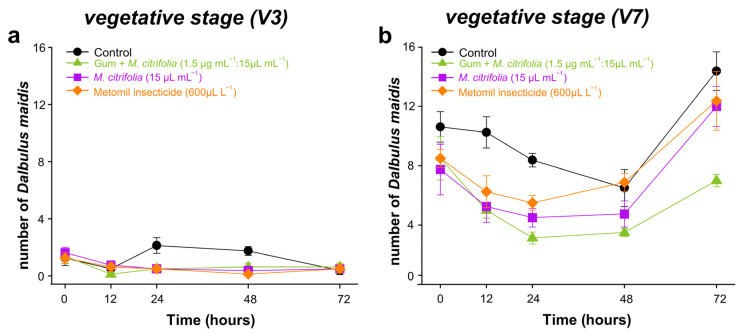
The effects of different insecticide solutions on the number of *Dalbulus maidis* adults infesting maize plants at vegetative stage V3 (**a**) and V7 (**b**). The symbol shows the average (±SE) of eight replicates.

**Table 1 biology-13-00479-t001:** *Morinda citrifolia* essential oil components were identified by gas chromatography coupled with mass spectrometry (GC-MS).

Chemical Class	Compound Name (IUPAC)	%	R.T. (min)	I.T.
Fatty acid	Octanoic acid 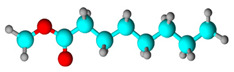	58.43	14.239	12.78
Fatty acid	Hexanoic acid 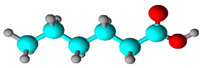	9.46	8.435	7.81
Fatty acid esters	Pent-4-enyl hexanoate 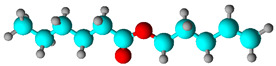	8.17	14.449	14.35
Fatty acid methyl ester	Methyl octanoate 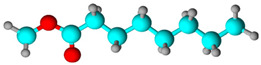	7.26	10.606	10.52

% Peak Area, R.T. Retention time, I.T. Retention index. In structural molecules, blue represents the element carbon, red represents the element oxygen, and gray represents hydrogen.

**Table 2 biology-13-00479-t002:** Velocity mycelium growth (mm) of *Curvularia lunata* over time when treated with *Morinda citrifolia* essential oil, angico gum (*Anadenanthera colubrina)*, and emulsion (angico gum + *M citrifolia*).

Treatment	Velocity of Growth Mycelial (mm/day ± SE) *Curvularia lunata*					VCM (mm/day)
	Day 2	Day 4	Day 6	Day 8	Day 10	
Control	18.2 ± 0.3	8.7 ± 0.5	8.2 ± 0.2	0.0 ± 0	0.0 ± 0	11.67
*Morinda citrifolia*	3.0 ± 0.3	4.2 ± 0.3	6.4 ± 0.5	3.2 ± 1.0	2.1 ± 2.1	3.78
angico gum	14.0 ± 0.9	2.0 ± 0.7	16.4 ± 0.4	2.6 ± 1.3	0.0 ± 0	8.75
Angico gum + *M. citrifolia*	1.9 ± 1.1	1.3 ± 0.7	4.8 ± 2.5	3.6 ± 0.6	3.5 ± 0	3.00

VCM: Velocity of mycelial growth.

## Data Availability

Data are contained within the article and Appendix A.

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
