# Peer review of "Morinda citrifolia Essential Oil: A Plant Resistance Biostimulant and a Sustainable Alternative for Controlling Phytopathogens and Insect Pests"

_biology, 2024, doi:10.3390/biology13070479_

Round 1
Reviewer 1 Report
Comments and Suggestions for Authors
The reviewer has read with interest the manuscript entitled as “Morinda citrifolia essential oil: A plant resistance biostimulant and a sustainable alternative for controlling phytopathogens and insect pests.” submitted by Bruna Leticia Dias et al. to the science journal “Biology”. Under the circumstance of public interest in environmental protection, the authors attempted to assess the potential of usage of essential oil of Morinda citrifolia and angico gum (Anadenanthera colubrina), instead of agricultural chemicals. As a result, they indicated that these substances of biological originx are proven to be effective as a fungicide and an insecticide and to become a sustainable biorational alternative. The manuscript is written passably but there are some serious problems as follows.
1. The deviations of data and significant differences are not shown in some of Figures and Tables. Thus, the readers cannot estimate existence or non-existence of significant differences among data. The existence or non-existence of statistical differences should be shown between comparable data.
2. There are many careless mistakes in spellings and grammatical usage. Some of them are pointed in the following list.
3. The authors used large volume for the discussion of the biochemical mechanisms of agricultural chemicals. The authors investigated mainly the agricultural effects of usage of substances of biological originx mainly by agricultural experiments but they examined only to some extent physiological and biochemical mechanism of these chemicals in this study. Thus, as the large volume of discussions on them are not suitable in this manuscript, please shorten these parts in Discussion.
Some problems are listed below and some suggestions are appended.
Line 45 Morinda citrifolia essential oil to induce plant resistance, and for controlling phytopathogens -> Morinda citrifolia essential oil to induce plant resistance, and to control phytopathogens
Line 54 It also, the -> The
Line 55 suggest -> suggested
Lines 57-59 The essential reduced in 76% the population of D. maidis adults, and repelled, under field conditions, in 50% the number of D. maidis after 48 hours. -> The essential oil reduced 76% of the population of D. maidis adults, and repelled 50% of the number of D. maidis after 48 hours under field conditions.
Line 60 by 50%The -> by 50%. The
Line 71 is among the -> is one of
Line 79 in conventional -> conventionally
Line 81 are -> have been
Lines 86-90 Additionally, M. citrifolia is adapted to withstand severe weather and soil conditions, is used in traditional medicine, and has been shown selective to predator insects, it could be considered, as a raw material economic source, safe to mammals and environmentally sustainable [10,16,17]. -> (This sentence is not grammatical. Please rewrite it.)
Lines 91-93 However, essential oils and their compounds, despite being toxic to insects and rapidly degrading in the environment, on the other hand, their persistence in the field is so limited that they often have almost no residual toxic action against pests [18]. -> (This sentence is not grammatical. Please rewrite it.)
Lines 98-101 So, the Anadenanthera colubrina (Fabaceae), popularly known as white angico is cultivated mainly in the Cerrado region of Brazil and produces a gum on the tree trunk and branches is very used as medicine by the population where occurs [21]. -> (This sentence is not grammatical. Please rewrite it.)
Lines 101-103 This gum is a complex polysaccharide, soluble in water in compounds by arabinose, mannose, galactose, glucopyranose, rhamnose, and hexuronic acid [22,23]. -> (This sentence is not grammatical. Please rewrite it.)
Line 108 pathogen-contained reactions -> (Please explain this phrase simply.)
Line112 By the exposed -> (The reviewer cannot understand this phrase.)
Lines 168-170 Homogenized with potassium phosphate buffer (100 mM; 7.0 pH), added EDTA (1 mM) and ascorbic acid (1mM). -> (This is not sentence. Please rewrite it.)
Line 173 adapted -> (Omit this word.)
Lines 179-181 Determined by the decomposition of H2O2, the activities of the enzymes catalase (CAT), phenol peroxidase (POX), and ascorbate peroxidase (APX) were monitored by the decrease in absorbance monitored for 3 minutes with readings every 15 seconds. -> (This sentence is not good. Please rewrite it.)
Line 185 H2O2 g-1 E-1 min-1). POX activity analyzed -> H2O2 (g-1 E-1 min-1). POX activity was analyzed
Line 190 The APX enzyme determined -> The APX enzyme activity determined
Line 191 using the adapted method described by described by [36] -> using the method described by Asada [36]
Line 200 was obtained at 550 nm -> was monitored at 550 nm
Lines 208-209 Curvularia lunata (Cochliobolus sativus sequence was used because of its high similarity - M2TAQ3) -> (Why did the authors not use Curvularia lunata? Please show the reason.)
Line 222 following [43]. -> following to Moura et al [43]
Line 228 Schrodinger, 2018 -> (This is not listed in References. Add the paper into References.)
Line 261 200 μL of treatments were -> (Show concretely the treatments.)
Line 270 described by [50] -> described by Maia et al [50]
Lines 277-278 oviposition. Adult insects were subsequently collected for laying for 4 to 6 days, with a small number of insects per greenhouse. -> (This sentence feels wrong. Please rewrite this sentence.)
Line 326 Table 1 (The values in Table 1 does not accurately correspond to those described in text. Please make the descriptions in text correspond to Table 1.)
Lines 347-349 Was evident that the POX activity was determined by the presence of C. lunata fungi, in the absence of these was no variation in the activity of POX that was higher in octanoic acid than M. citrifolia (Figure 1d). -> (The reviewer cannot understand this sentence. Please rewrite it.)
Line 352 increase only was only in intermediary concentration -> increase only in intermediary concentration
Line 356 Figure 1 (Because standard deviations are not shown in most of measured values in Figure 1, it is not clear whether or not the measured points are significantly different each other and whether or not the regression lines are adequate. Please reconsider the issues.)
Lines 364 of M. citrifolia essential oil, and angico gum is shown in Figure 2. -> of M. citrifolia essential oil and angico gum is shown in Figure 2.
Lines 364-367 In addition, were not observed symptoms of phytotoxicity with high levels of chlorosis and necrosis in the plants treated with the lethal doses (both treatments) to D. maidis, so it can be considered safe for maize plants. -> (This sentence is not perfect. Please rewrite this.)
Lines 371-397 3.4. Molecular docking -> (As the reviewer has no experiences on these researches, he cannot evaluate these results.)
Line 401 the (AUDPC) in increases, however -> the AUDPC increases but
Line 407 Figure 4 The vertical axis label: Are Under the Disease -> Area Under the Disease
(Please show the deviations of each measured values.)
Line 423, Line 435 Figures 5&6 (Standard deviations are shown in some data and not shown in other data. How did the authors estimate statistical differences between these data?)
Lines 440-442 Our results show a significant effect of the insecticide, vegetative stage and the interaction these on the repellence of vector D. maidis; similarly, the time and their interaction with the vegetative stage also was evidents (Table S4). -> (This sentence is not good. Please rewrite this.)
Line 446 Figure 7 (The vertical axis label: What is (n○)?
Comments on the Quality of English Language
There are many careless mistakes in spellings and grammatical usage. Some of them are pointed in the comments to the authors.
Author Response
June 09th , 2024
Dear Review
We are pleased to re-submit our manuscript biology-3041344 entitled “Morinda citrifolia essential oil: A plant resistance biostimulant and a sustainable alternative for controlling phytopathogens and insect pests” to the Journal Biology. First of all, we would like to thank you for reviewing our manuscript and pointing out areas for scientific improvement. We have gladly taken them all into account in the revised version of our manuscript. To make it easier to check the changes, the lines of the manuscript have been numbered and all changes have been marked in red in the body of the manuscript for your information.
Sincerely,
Prof. Dr. Gil Rodrígues dos Santos
Production Vegetal Pos-graduate Program
Federal University of Tocantins
Tocantins – Brazil

Reviewer 2 Report
Comments and Suggestions for Authors
Essential oils have a wide range of applications for preventing and controlling crop diseases and pests. Especially, morinda citrifolia essential oil is a renewable plant resource with the function of protecting the stomach and improving sleep. Bruna Leticia Dias et al used Morinda citrifolia essential oil to develop a new biostimulant and biological control agent. Thereby, it is an effective biopesticide for controlling crop diseases and insect pests. It can be seen that Morinda citrifolia essential oil is an effective and sustainable reliable alternative to chemical pesticides. Nevertheless, the study still has some flaws in its current form. Therefore, this manuscript needs major revisions before possible publication in Biology.
The comments are provided below:
[1] For the Graphical Abstract, “Up” or “down”-arrows next to the “SDO, CAT, APX, POX, and QUIT” should be provided. These arrows are meant to illustrate that the “SDO, CAT, APX, POX, and QUIT” are up-regulation or down-regulation.
[2] In the revised version, the number of decimals is kept consistent in the whole text. The sections of 2.3, 2.4, 2.5, 2.7, and etc. Such as, “3.0, 9.0, 15, 20, and 30 µL mL-1”, “0.3, 0.9, 1.5, 2.0, and 3.0 µg mL-1”, and “6.25, 12.5, 25, 50, and 75 µL mL-1”. So, it’s better to unify the number of decimals in the whole text.
[3] In the section “2.4. Phytotoxicity of M. citrifolia and Anadenanthera colubrina in corn leaf”, especially for the “0%”, authors should directly use number instead of percentages.
[4] In the section “2.6.3. Molecular docking calculations and 2.6.4. Molecular dynamics simulations”, a detailed molecular docking process containing the homologous modeling rating situation should be provided in the revised version.
[5] In the section “2.9.1. Toxicity”, the statement that “Five adult insects were collected” is not clear. The key process of collecting insects should be detailedly described in the main text.
[6] In the part of “2.10. Statistical analysis”, what is the number of independent replicates for each experiment?
[7] In Figure 2, at least three pictures are provided for every treatment in the revised version. Especially, for the different control treatments of Anadenanthera colubrina and Morinda citrifolia treatments, different pictures must be provided in the revised version.
[8] A bias value has been added for Figure 4 in the revised version.
[9] In Figures 2 and 4, the unit of drug concentration was μg mL-1, but the unit of drug concentration was mg mL-1 in Figure 1. Please unify the unit (μg mL-1 or mg mL-1) in the revised version. This will reduce confusion when reading the manuscript.
Comments on the Quality of English Language
Minor editing for the English language.
Author Response
June 9th, 2024
Dear Review
We are pleased to re-submit our manuscript biology-3041344 entitled “Morinda citrifolia essential oil: A plant resistance biostimulant and a sustainable alternative for controlling phytopathogens and insect pests” to the Journal Biology. First of all, we would like to thank you for reviewing our manuscript and pointing out areas for scientific improvement. We have gladly taken them all into account in the revised version of our manuscript. To make it easier to check the changes, the lines of the manuscript have been numbered and all changes have been marked in red in the body of the manuscript for your information.
Sincerely,
Prof. Dr. Gil Rodrígues dos Santos
Production Vegetal Pos-graduate Program
Federal University of Tocantins
Tocantins – Brazil

Round 2
Reviewer 1 Report
Comments and Suggestions for Authors
The reviewer has read the revised manuscript entitled as “Morinda citrifolia essential oil: A plant resistance biostimulant and a sustainable alternative for controlling phytopathogens and insect pests.” submitted by Bruna Leticia Dias et al. to the science journal “Biology”. The manuscript has been revised well but some problems remains as follows. The authors should revise again the manuscript at full stretch.
Line 71 is one the -> is one of the
Line 88 considered an affordable source of raw material for several uses. -> considered as an affordable source of raw material for several uses.
Line 106 pathogen-related reactions similar to those direct -> pathogen-related reactions similar to those directly
Line 116 we were able to predict potential interactions for the M. citrifolia -> we were able to predict potential interactions of the M. citrifolia
Line 165 the parcel of plants (pathogen) was inoculated with -> the parcel of plants (containing pathogen) was inoculated with
Line 180 peroxidase (APX) were monitored by the decrease in absorbance > peroxidase (APX) were measured by the decrease in absorbance
Line 224 PyMOL 2.0 [45] and > (Though the reviewer previously pointed the reference of Schrodinger, it was not good. The reference is not necessary and only PyMOL 2.0 is OK with no references. Please omit the reference [45].
Lines 260-262 Subsequently, in separate Petri dishes, 200 μL of each solution (i.e., essential oil, angico gum, and essential oil + angico gum) were distributed on the surface of the commercial PDA culture medium (Potato-Dextrose-Agar) with a Drigalsky loop. > Subsequently, 200 μL of each solution (i.e., essential oil, angico gum, and essential oil + angico gum) were distributed on the surface of the commercial PDA culture medium (Potato-Dextrose-Agar) in separate Petri dishes with a Drigalski loop.
Lines 280-284 an aspirator made with a falcon tube and two 13 mm diameter hoses. One of the hoses, covered with voile fabric, provided suction, while the other, a longer hose, was positioned over the insect, which was then sucked into de falcon tube. > (The reviewer guesses that the apparatus which the authors describe here is an insect suction pipe. If so, any detailed explanations are not necessary.)
Line 283 de falcon tube > falcon tube
Line 324 hexanoic acid (9,46%), pent-4-enyl hexanoate (8,17%) > hexanoic acid (9.46%), pent-4-enyl hexanoate (8.17%)
Line 352 independent of the treatment involving -> independent of the treatments involving
Lime 354 showing a negative correlation for octanoic acid and a positive correlation for pure essential oil. > showing a negative correlation to octanoic acid and a positive correlation to pure essential oil.
Lines 383-385 Hexanoic acid (-382 4.2 Kcal mol-1), 4-pentenyl ester [pent-4-enyl hexanoate] (-4.6 Kcal mol-1), octanoic acid, methyl ester [methyl octanoate] (-4.4 Kcal mol-1), hexanoic acid, methyl ester [methyl hexanoate] (-4.1 Kcal mol-1) and isobutyl 3-methylbut-3-enyl carbonate [3-methylbut-3-enyl 2-methyl propyl carbonate] (-4.6 Kcal mol-1). > (Only the chemicals are simply listed without any explanations. Hexanoic acid appears twice.)
Line 406 (AUDPC) increases > (Are the parentheses necessary?)
Line 447 insecticide solution, the vegetative stage of the plant, and by the interaction > insecticide solution, the vegetative stage of the plant, and the interaction
Lines 491-492 An analysis of enzymes and their mode of action in healthy plants and those infected with Curvularia lunata provides insights into the pathogen-host interaction. > The analyses of enzymes and their mode of action in healthy plants and infected plants with Curvularia lunata provide insights into the pathogen-host interaction.
Lines 493—496 Superoxide dismutase (SOD) is activated in the presence of the pathogen, and as a resistance activator, the compounds before fungal infection substantially increase the amount of the enzyme, pre- fungal infection treatment substantially increases the enzyme’s quantity, accelerating the attack on the pathogen. > (This part is difficult to understand the contents. What is ‘the compounds before fungal infection’? Please show the compounds in the concrete.
Line 497 which degrades H2O2 without a reducing agent > (What is ‘a reducing agent’?)
Line 502 Cymbopogon citratus increased resistance against Pseudomonas syringae -> Cymbopogon citratus increases resistance against Pseudomonas syringae
Lines 503-504 these levels vary according to stresses > (Do the levels mean the response strengths? If so, please write them concretely.)
Comments on the Quality of English Language
The English of the manuscript contains not a little careless and grammatical mistakes in the first manuscript and the revised one. Probably the first revision of the manuscript was done in solitude. The authors should revise the manuscript with the coauthors and should not revise it alone.
Author Response
June 15, 2024
Dear Review
We are pleased to re-submit our manuscript biology-3041344 entitled “Morinda citrifolia essential oil: A plant resistance biostimulant and a sustainable alternative for controlling phytopathogens and insect pests” to the Journal Biology. First of all, we would like to thank you for reviewing our manuscript and pointing out areas for scientific improvement. We have gladly taken them all into account in the revised version of our manuscript. To make it easier to check the changes, the lines of the manuscript have been numbered and all changes have been marked in red in the body of the manuscript for your information. In addition, we inform you that the English has been improved by a professional and has also been checked by the software (www.grammarly.com) to address the concerns raised.
[Line 71] is one the -> is one of the
Reply: Done as suggested! (see line 71)
[Line 88] considered an affordable source of raw material for several uses. -> considered as an affordable source of raw material for several uses.
Reply: Done as suggested! (see line 88)
[Line 106] pathogen-related reactions similar to those direct -> pathogen-related reactions similar to those directly
Reply: Done as suggested! (see line 106)
[Line 116] we were able to predict potential interactions for the M. citrifolia -> we were able to predict potential interactions of the M. citrifolia
Reply: Done as suggested! (see line 116)
[Line 165] the parcel of plants (pathogen) was inoculated with -> the parcel of plants (containing pathogen) was inoculated with
Reply: Done as suggested! (see line 165)
[Line 180] peroxidase (APX) were monitored by the decrease in absorbance > peroxidase (APX) were measured by the decrease in absorbance
Reply: Done as suggested! (see line 180)
[Line 224] PyMOL 2.0 [45] and > (Though the reviewer previously pointed the reference of Schrodinger, it was not good. The reference is not necessary and only PyMOL 2.0 is OK with no references. Please omit the reference [45].
Reply: Done as suggested! (see line 224)
[Lines 260-262] Subsequently, in separate Petri dishes, 200 μL of each solution (i.e., essential oil, angico gum, and essential oil + angico gum) were distributed on the surface of the commercial PDA culture medium (Potato-Dextrose-Agar) with a Drigalsky loop. > Subsequently, 200 μL of each solution (i.e., essential oil, angico gum, and essential oil + angico gum) were distributed on the surface of the commercial PDA culture medium (Potato-Dextrose-Agar) in separate Petri dishes with a Drigalski loop.
Reply: Done as suggested (see line 260-262)
[Lines 280-284] an aspirator made with a falcon tube and two 13 mm diameter hoses. One of the hoses, covered with voile fabric, provided suction, while the other, a longer hose, was positioned over the insect, which was then sucked into de falcon tube. > (The reviewer guesses that the apparatus which the authors describe here is an insect suction pipe. If so, any detailed explanations are not necessary.)
Reply: That’s right the apparatus is an insect suction pipe, We, have rewritten the sentences to make them clearer and more concise (see lines 280-281)
[Line 283] de falcon tube > falcon tube
Reply: We, have rewritten the sentences to make them clearer and more concise. (see line 280-281)
[Line 324] hexanoic acid (9,46%), pent-4-enyl hexanoate (8,17%) > hexanoic acid (9.46%), pent-4-enyl hexanoate (8.17%)
Reply: Done as suggested (see line 321)
[Line 352] independent of the treatment involving -> independent of the treatments involving
Reply: Done as suggested (see line 349)
[Lime 354] showing a negative correlation for octanoic acid and a positive correlation for pure essential oil. > showing a negative correlation to octanoic acid and a positive correlation to pure essential oil.
Reply: Done as suggested (see line 351,352)
[Lines 383-385] Hexanoic acid (-382 4.2 Kcal mol-1), 4-pentenyl ester [pent-4-enyl hexanoate] (-4.6 Kcal mol-1), octanoic acid, methyl ester [methyl octanoate] (-4.4 Kcal mol-1), hexanoic acid, methyl ester [methyl hexanoate] (-4.1 Kcal mol-1) and isobutyl 3-methylbut-3-enyl carbonate [3-methylbut-3-enyl 2-methyl propyl carbonate] (-4.6 Kcal mol-1). > (Only the chemicals are simply listed without any explanations. Hexanoic acid appears twice.)
Reply: Done as suggested (see line 379)
[Line 406] (AUDPC) increases > (Are the parentheses necessary?)
Reply: Done as suggested (see line 401)
[Line 447] insecticide solution, the vegetative stage of the plant, and by the interaction > insecticide solution, the vegetative stage of the plant, and the interaction
Reply: Done as suggested (see line 444)
[Lines 491-492] An analysis of enzymes and their mode of action in healthy plants and those infected with Curvularia lunata provides insights into the pathogen-host interaction. > The analyses of enzymes and their mode of action in healthy plants and infected plants with Curvularia lunata provide insights into the pathogen-host interaction.
Reply: Done as suggested (See line 488,489)
[Lines 493—496] Superoxide dismutase (SOD) is activated in the presence of the pathogen, and as a resistance activator, the compounds before fungal infection substantially increase the amount of the enzyme, pre- fungal infection treatment substantially increases the enzyme’s quantity, accelerating the attack on the pathogen. > (This part is difficult to understand the contents. What is ‘the compounds before fungal infection’? Please show the compounds in the concrete.
Reply: We rewrote the sentences (see line 490-493)
[Line 497] which degrades H2O2 without a reducing agent > (What is ‘a reducing agent’?)
Reply: We have rewritten the sentences to make them clearer and more concise. See line 494-496
[Line 502] Cymbopogon citratus increased resistance against Pseudomonas syringae -> Cymbopogon citratus increases resistance against Pseudomonas syringae
Reply: Done as suggested (see line 499-500)
[Lines 503-504] these levels vary according to stresses > (Do the levels mean the response strengths? If so, please write them concretely.)
Reply: We have rewritten the sentences to make them clearer and more concise. See line 500-502
Comments on the Quality of English Language
The English of the manuscript contains not a little careless and grammatical mistakes in the first manuscript and the revised one. Probably the first revision of the manuscript was done in solitude. The authors should revise the manuscript with the coauthors and should not revise it alone.
Reply: We inform you that the English has been improved by a professional and has also been checked by the software (www.grammarly.com) to address the concerns raised.
Sincerely,
Prof. Dr. Gil Rodrígues dos Santos
Production Vegetal Pos-graduate Program
Federal University of Tocantins
Tocantins – Brazil
Reviewer 2 Report
Comments and Suggestions for Authors
[3] In the section “2.4. Phytotoxicity of M. citrifolia and Anadenanthera colubrina in corn leaf”, especially for the “0%”, authors should directly use numbers instead of percentages. Change “0%” to “0”, other changes are unnecessary.
[7] In Figure 2, at least three pictures are provided for every treatment in the revised version. Especially, for the different control treatments of Anadenanthera colubrina and Morinda citrifolia treatments, different pictures must be provided in the revised version. Please supplement experiments according to the method in Figure 11 [Int. J. Mol. Sci. 2023, 24, 2897. https://doi.org/10.3390/ijms24032897]. Three leaves of each treatment are exhibited in revised Figure 2.
Author Response
June 15, 2024
Dear Review
We are pleased to re-submit our manuscript biology-3041344 entitled “Morinda citrifolia essential oil: A plant resistance biostimulant and a sustainable alternative for controlling phytopathogens and insect pests” to the Journal Biology. First of all, we would like to thank you for reviewing our manuscript and pointing out areas for scientific improvement. We have gladly taken them all into account in the revised version of our manuscript. To make it easier to check the changes, the lines of the manuscript have been numbered and all changes have been marked in red in the body of the manuscript for your information.
[3] In the section “2.4. Phytotoxicity of M. citrifolia and Anadenanthera colubrina in corn leaf”, especially for the “0%”, authors should directly use numbers instead of percentages. Change “0%” to “0”, other changes are unnecessary.
Reply: Done as suggested! (see line 156)
[7] In Figure 2, at least three pictures are provided for every treatment in the revised version. Especially, for the different control treatments of Anadenanthera colubrina and Morinda citrifolia treatments, different pictures must be provided in the revised version. Please supplement experiments according to the method in Figure 11 [Int. J. Mol. Sci. 2023, 24, 2897. https://doi.org/10.3390/ijms24032897]. Three leaves of each treatment are exhibited in revised Figure 2.
Reply: Done as suggested! (see figure 2)
Sincerely,
Prof. Dr. Gil Rodrígues dos Santos
Production Vegetal Pos-graduate Program
Federal University of Tocantins
Tocantins – Brazil